# Colonoscopy: Preparation and Potential Complications

**DOI:** 10.3390/diagnostics12030747

**Published:** 2022-03-18

**Authors:** Wojciech Latos, David Aebisher, Magdalena Latos, Magdalena Krupka-Olek, Klaudia Dynarowicz, Ewa Chodurek, Grzegorz Cieślar, Aleksandra Kawczyk-Krupka

**Affiliations:** 1Center for Laser Diagnostics and Therapy, Department of Internal Diseases, Angiology and Physical Medicine, Specialist Hospital No. 2, 41-902 Bytom, Poland; wojlatos@gmail.com; 2Department of Photomedicine and Physical Chemistry, Medical College of Rzeszów University, University of Rzeszów, 35-959 Rzeszów, Poland; daebisher@ur.edu.pl; 3Silesian Centre for Heart Disease in Zabrze, Marii Curie Skłodowskiej 9, 41-800 Zabrze, Poland; latos.magdalena93@gmail.com; 4Center for Laser Diagnostics and Therapy, Department of Internal Medicine, Angiology and Physical Medicine, Medical University of Silesia in Katowice, 41-902 Bytom, Poland; magda.krupka94@gmail.com (M.K.-O.); cieslar1@tlen.pl (G.C.); 5Center for Innovative Research in Medical and Natural Sciences, Medical College of the University of Rzeszów, 35-310 Rzeszów, Poland; kdynarowicz@ur.edu.pl; 6Department of Biopharmacy, Faculty of Pharmaceutical Sciences in Sosnowiec, Medical University of Silesia in Katowice, 41-208 Sosnowiec, Poland; echodurek@sum.edu.pl

**Keywords:** colonoscopy, gastroenterology, diagnostics

## Abstract

Colonoscopy is a fairly common test that serves both diagnostic and therapeutic purposes. It has been considered the gold standard in colorectal cancer screening for several years. Due to the nature of the examination, various types of complications may occur. The purpose of this analysis is to describe the various complications related to the period of preparation for colonoscopy among hospitalized patients, including life-threatening ones, in order to know how to avoid complications while preparing for a colonoscopy. We analyzed the nursing and medical reports of 9962 patients who were prepared for colonoscopy between 2005 and 2016. The frequency of various side effects associated with intensive bowel cleansing prior to colonoscopy was assessed. In justified cases, additional medical data were collected from patients, their families or from other doctors providing advice to patients after complications. Out of 9962 patients prepared for colonoscopy, 180 procedures were discontinued due to complications and side effects, and in these cases no colonoscopy was performed. The most common complications were: vomiting; epistaxis; loss of consciousness with head injury; abdominal pain; acute diarrhea; symptoms of choking; heart rhythm disturbances; dyspnea; fractures of limbs and hands; acute coronary syndrome; hypotension; hypertension; cerebral ischemia; severe blood glucose fluctuations; increased muscle contraction and allergic reactions. In addition to the documentation of our own research, several works of other research groups were also analyzed. Currently, the literature does not provide data on the frequency and type of complications in the preparation period for colonoscopy. The advantage of our work is the awareness of the possibility of serious complications and postulating the necessary identification of threats. Individualization of the recommended procedures and increased supervision of patients undergoing bowel cleansing procedure, we hope, will reduce the occurrence of complications and side effects.

## 1. Introduction

Colonoscopy is a widely used method of diagnosing and treating gastrointestinal disease [1,2]. Endoscopic examination of the lower gastrointestinal tract, called colonoscopy, was, is and will remain the ‘gold standard’ in the diagnosis and treatment of majority of diseases of the large intestine [3,4,5,6,7,8,9,10]. The constantly growing number of cases of various types of intestinal diseases is associated with an increased demand and availability of colonoscopic examinations. New means of cleansing the intestine, as well as patterns and ways of administering those procedures, are still appearing [11,12,13,14,15,16,17,18,19,20].

Diagnostic colonoscopy and, to a greater extent, interventional colonoscopy are subject to possible adverse events, including severe difficulties that significantly threaten the life and health of patients. Appropriate bowel cleansing is an indispensable requirement for a well-performed and planned endoscopic procedure [21,22,23,24,25,26,27]. However, in an ageing society the possibility of various comorbidities in patients qualified for endoscopic examinations and procedures is much higher. Preparation for colonoscopy is a procedure stretched over time, performed for many hours, and largely unpredictable as to its individual course. Macrogols (polyethylene glycol (PEG)) are most frequently used as the means to prepare for diagnostic endoscopic and radiological tests or for operations on the large intestine [28,29,30,31]. Sufficiently effective cleansing of the intestine before endoscopy is a necessary condition, although it leads to multidirectional disturbances, mainly related to water and electrolyte levels. Until 2009, patients were usually prepared for examination one day before the procedure. Patients undergoing colonoscopy may be divided into two large groups. The first one consists of patients who prepare themselves at home. They follow recommendations of the physician who referred or qualified them for colonoscopy as part of their treatment or for diagnostic purposes. The other group are patients who are hospitalized for various reasons, suffering from diseases and ageing. They are prepared for an examination or an endoscopic procedure in a hospital ward, under medical and nursing supervision. The most common complications during preparation for a colonoscopy and after the examination include: vomiting; colicky stomach ache; dizziness and balance disorders; impaired consciousness with head injury; bone fractures and hemorrhages and epistaxis.

The subject of this analysis is the incidence of various adverse events, including life-threatening ones, related to the period of preparation for colonoscopy among hospitalized patients. This is to reduce (limit) the incidence of complications in preparation for a colonoscopy.

## 2. Materials and Methods

The Specialist Hospital No. 2 in Bytom houses the Central Endoscopy Laboratory, where endoscopic examinations are performed on patients who were hospitalized and outpatient visitors to the clinic. In the years 2005–2016, a total of 13,759 diagnostic and endoscopic procedures were performed within the lower gastrointestinal tract in adults between 18 and 93 years of age.

The Department of Internal Medicine, Angiology and Physical Medicine, Medical University of Silesia in Bytom, is a multi-profile, 75-bed internal disease department. In a time-span of 12 years (from 1 January 2005 to 31 December 2016), 9962 endoscopic procedures of the lower gastrointestinal tract were performed which required preparation of patients who were hospitalized in the clinic.

Out of the cases included in the analysis, 180 were cases of incomplete colonoscopy procedures, while the number 9782 was the number of colonoscopy procedures performed to the end. Figure 1 shows a diagram of procedures included in the analysis.

The quantity and percentage of non-fully successful bowel preparation before colonoscopy are presented on Figure 2. The pattern of administration of Fortrans in divided doses: two on the day before and two in the early morning hours on the day of the examination, dominated the procedure. For two consecutive years, since 2011, all patients without exception were prepared with the use of Moviprep.

Colonoscopy examinations, both diagnostic and therapeutic, were performed using Olympus endoscopic sets. Analysis of the causes of 180 unsuccessful colonoscopies, including the period of bowel preparation, is presented in Figure 3. The data regarding the occurrence of adverse events during the preparation period for colonoscopy were obtained from nursing and medical reports. In justified cases, additional medical data was collected personally or by phone, obtaining information from patients, their families or from other doctors.

All cases resulting in adverse effects of the preparation period for colonoscopy have been carefully analyzed, with particular emphasis on the serious adverse events threatening the life and health of patients.

At the same time, the frequency of occurrence of various adverse effects associated with intensive intestinal cleansing before colonoscopy was estimated. Emphasis was put on the frequency of their occurrence in hospitalized patients pre-qualified for endoscopic examinations, such as people burdened with metabolic, cardiac, pulmonary, neurological diseases and with age.

Intake of preparations which intensely cleanse the intestines in the process of colonoscopy preparation is almost always accompanied by varying degrees of dyspeptic symptoms, discomfort and abdominal pain.

In this study, the incidence of trivial symptoms, such as discomfort; nausea; feelings of fullness and bloating or transient moderate abdominal pain, was not taken into consideration. The analysis included only those events and side effects which were reflected in nursing reports and medical records.

## 3. Results

Figure 4 presents the prevalence of adverse effects occurring during the analysis of 9962 initiated bowel preparation regimens that required medical intervention at the ward, and/or implementation of specialized medical procedures.

The most common undesirable symptom of the bowel cleansing period in patients undergoing colonoscopy was severe vomiting that required medical attention. This was reported in 667 patients. It accounted for nearly 6.7% of all the intestinal cleansing procedures started. In 14% of cases, vomiting was mixed with blood. Urgent gastroscopy was prescribed in 75% of the cases.

Nose bleeding occurred in only 0.6% of all the subjects who underwent bowel cleansing. Self-help and nursing help turned out to be sufficient. Only six patients required a nasal tamponade. Weakness was found in 8% of cases (555 patients) undergoing colon cleansing, and in 1.49% of cases it was accompanied by fainting. Cases of unconsciousness were also reported. In the event of fainting and loss of consciousness, a total of 56 cases of head trauma were reported, which accounted for approximately 0.6% of the analyzed patients undergoing colonoscopy. The third most frequent occurrence was symptomatic abdominal pain of an intensity requiring administration of analgesics and antispasmodics. This was recorded in 496 cases. which accounted for nearly 6% of all patients. Only slightly more rarely, in 403 cases (5.8%), the need to administer painkillers was reported due to severe headache complaints reported by the patients.

Intake of laxatives causing acute diarrhea resulted in 54 (0.77%) cases of acute bleeding from the hemorrhoids. In 16 cases they were reported as hemorrhage and in three cases urgent surgical procedures were required. There was also one case of acute rectal prolapse.

Among the patients referred for a diagnostic colonoscopy for detection of gastrointestinal tract cancer, there were 14 cases of sub-obstruction, and four cases of full-blown obstruction during the preparation for the examination. In all 18 cases, which concerned 0.26% of the commenced cleaning procedures, general abdominal X-ray examinations were performed for diagnostic purposes. In cases of sub-obstruction, the symptoms subsided after conservative pharmacological treatment, and in cases of full obstruction, the patients were operated on as a matter of urgency.

In 3.7% of cases (258 patients), heart rhythm disturbances were reported by the patients. The feeling of palpitation or uneven rhythm was recorded 130 times. A total of 115 patients reported a too rapid heartbeat, and 13 had the feeling of an excessively slow rhythm of their heartbeat.

In 145 cases patients reporting weakness complained of dyspnea. In 1.7% (116) of cases this was accompanied by retrosternal pain which appeared during the colonoscopy preparation procedure. During the colonoscopy bowel cleansing 723 urgent ECG examinations were performed, though possibly not only because of the complaints reported by the patients. That number constitutes more than 10% of the intestinal cleansing procedures. Eventually, in the said group of patients the following diagnoses were temporarily related to the intestinal cleansing procedure: 138 additional supraventricular spasms; 35 patients with supraventricular tachycardia; 7 instances of atrial fibrillation occurring for the first time and 15 cases of consecutive paroxysmal atrial fibrillation. There was one incidence of a sinus arrest, one case of ventricular tachycardia and two patients suffered from ventricular bigeminy.

A total of 19 cases of acute coronary syndromes were reported. In seven patients, acute coronary syndrome occurred for the first time and in another 12 patients it was an exacerbation of the previously diagnosed chronic coronary disease. Moreover, there was one case of a sudden cardiac arrest, followed by an effective resuscitation. In all, 13 patients were urgently referred for an emergency coronarography and, in nine cases, stents were implanted after a diagnosis of critical coronary stenosis.

## 4. Discussion

The analysis of the results of studies in which severe vomiting occurred showed that the most common cause of blood in vomiting was Mallory–Weiss syndrome (25 cases); gastritis (6 cases); active duodenal ulcer (6 cases) and stomach cancer (5 cases). Emergency gastroscopy, due to bleeding from the upper gastrointestinal tract during bowel cleansing, made it possible to diagnose stomach cancer. In 12 cases of emergency gastroscopy, no upper gastrointestinal bleeding was found despite the presence of blood in the vomit (Forrest III). In two cases, the presence of blood in the vomit was caused by epistaxis, and in one case by bleeding from the gums.

X-rays of the skull were taken urgently in most cases of fainting and loss of consciousness resulting in a head injury. In three cases of fainting, they suffered nose fractures, in one of the zygomatic bone, and in another two broken teeth. In addition, two patients suffered a rib fracture as a result of a fall while fainting, and one patient suffered a rib fracture as a result of a fall and unconscious trauma. In three cases, upper limbs were fractured as a result of fainting and falling; two arm fractures and one Colles fracture at the distal end of the radius.

According to medical records, in 2.6% (178) cases of intestine cleansing, symptomatic cerebral circulatory disorders occurred. The patients suffered from dizziness with disturbances of balance. In 11 cases, neurological symptoms occurred that required these patients to be transferred to neurological wards. There, seven patients were diagnosed with transient ischemia of the brain (TIA), and there were three cases of ischemic stroke (two patients with speech aphasia and hemiparesis, and one patient with speech aphasia only). In the case of one patient, the diagnosed stroke was of a hemorrhagic nature and was accompanied by speech aphasia and hemiparesis. Among those hospitalized and prepared for colonoscopy, patients previously diagnosed with diabetes constituted a special group. Despite that, 117 cases of symptomatic hypoglycemia with necessary intravenous glucose administration were reported. That concerned, respectively, 8 patients with type I diabetes, 102 patients with type II diabetes and 7 patients with type III diabetes. At the same time, 27 cases of hypoglycemia not associated with previously diagnosed diabetes were revealed. In total, hypoglycemia was diagnosed in 144 patients undergoing preparation for colonoscopy. Higher blood glucose levels that required insulin or correction of current treatment were reported less frequently. High and significant hypoglycemia occurred in 75 patients, which concerned 2 patients with type I diabetes, 67 patients with type II diabetes and 6 patients with type III diabetes. In addition, nine patients showed abnormal blood glucose levels during preparation for intestinal cleaning or immediately before the colonoscopy. This, in turn, gave grounds to suspect and diagnose diabetes de novo. In total, 3.3% (228 patients) undergoing preparation for colonoscopy revealed significant glycemic disorders, requiring urgent intervention.

Apart from the occurrence of reported adverse events, it was found that a group of another 44 patients (0.6% of the prepared), suffered from increased muscle contractility after the use of intestinal cleansing products, with symptomatic tetany diagnosed in 11 cases. There were also 27 cases of pruritus with a skin rash of allergic nature, with variable intensity of changes/lesions. Those patients responded well to administration of anti-allergic drugs. In five cases, the skin symptoms took the form of general urticaria.

Widespread availability and thus frequency, of endoscopic diagnosis of the lower gastrointestinal tract is inextricably connected with a large number of adverse events. Other factors worth mentioning are: constant development of endoscopic methods of treatment and the wide variety of patients, including those who are of older age and with comorbidities. It is an established belief that the safety of colonoscopy depends on the quality and degree of intestinal cleaning. The awareness of possible adverse events related to the introduction of the endoscope into the intestine, manipulation within it, and the invasiveness of endoscopic procedures in general is quite high. The knowledge of the risks and scale of occurrence of adverse events in bowel preparation period for colonoscopy is less widespread.

No comprehensive data or full analyses in this field could be found in the ‘Pub Med’ database. Therefore, endoscopy operators and physicians who qualify patients for colonoscopy, and are responsible for the safety of patients under their care, should be familiar with the possible adverse events of the preparatory period, if only to know how to prevent and how to treat them. It is well known that the prevention of adverse events is better than treatment. Currently there are no ready-made procedures, nor sufficient guidelines or clear schemes for modifying the preparation methods prior to colonoscopy. Proper and safe preparation of the intestine is necessary for a successful colonoscopy; therefore, the used intestinal preparatory substance is the most important factor [30]. Before endoscopic intestinal examination became commonly performed, intestinal cleaning prior to radiological examinations and surgical procedures used to be performed using dietary restrictions, laxatives and enemas. That procedure had many side effects [32]. Along with the popularization of colonoscopy, many dedicated measures and their administration regimens were used to cleanse the intestine sufficiently.

The development and use of polyethylene glycol solutions (PEG) proved to be a significant advancement. According to Anastassopoulos et al., the cumulative incidence of adverse effects was significantly lower in the cohorts in which SUPREP^®^ (intra-oral sodium sulphate, magnesium sulphate and sodium sulphate) was administered, as compared to other commonly used formulas of intestinal preparations. Researchers concluded that sodium sulphate is just as safe, or perhaps even safer, than other preparations [33]. There are also commercially available solutions of sodium picosulfate and magnesium citrate, osmotically active preparations, stimulant laxatives, anti-emetic and prokinetic drugs [34,35].

Despite advances in the field, patients often think that preparation of the intestine is the most unpleasant part of the examination. In the published meta-analyses, there are no unequivocal conclusions as to which method provides the best cleansing [32,36]. Apart from the choice of a preparation, the influence of diet is also mentioned among the factors that improve the quality of cleaning. Recommendation of a purely liquid diet is probably irrelevant, but a low fiber regime may be more beneficial as it improves the patient’s satisfaction. Effective cleansing procedure for patients with acute bleeding from the lower gastrointestinal tract is a particularly difficult challenge [36]. Only a well-prepared intestine, cleaned from faeces and blood, gives good prospects of identifying the source of bleeding, defining treatment options and enabling endoscopic treatment [37,38]. In cases of acute bleeding from the lower gastrointestinal tract; cleansing of the intestine before colonoscopy potentially increases the risk of vomiting; aspiration pneumonia; volume overload and changes in vital signs associated with increased blood loss [39]. Moreover, perforation, cardiovascular events and sepsis were reported as adverse effects associated with the period of preparation for urgent colonoscopy [40].

Japanese researchers compared the number of adverse events in the intestinal cleansing period in patients who were prepared urgently due to acute bleeding from the lower gastrointestinal tract (9%) to those where no bleeding occurred (7%). No significant differences were discovered in the number of adverse events in both groups [41]. An example of a patient who developed severe symptomatic hyponatremia, after PEG was performed for a planned colonoscopy, showed that life-threatening water and electrolyte disorders may affect patients with risk factors such as older age, use of thiazide diuretics and SSRI, as well as those with renal or heart failure.

Patients with risk factors for colonoscopy should be closely monitored and treated promptly in the event of adverse events to avoid permanent neurological sequelae and death. In order to prevent severe adverse events, such as the osmotic demyelinating syndrome, it is particularly important to avoid rapid correction of sodium concentration in patients requiring treatment of hyponatremia [42]. The ideal preparation for bowel cleansing should be effective, safe and well tolerated by the patient [43,44,45,46,47,48,49,50].

## 5. Conclusions

There are currently no data in the literature on the frequency and type of adverse events of the preparatory period for colonoscopy. High frequency and variety of described adverse events related to intestinal cleaning and the possibility of severe and serious adverse events is the weight of the problem being raised. It is advisable to individualize qualifications as to the choice of means and methods of preparation, including necessity of hospitalization and monitoring its course. It is impossible to directly compare different types and ways of preparing patients for colonoscopy in terms of various adverse events due to a disproportionality between groups, as well as without an analysis of aggravating conditions.

The lack of reliable information and elementary cooperation on the part of the patient during the preparatory period for colonoscopy may nullify the effect of preparation and expose the patient to adverse events and increase their severity.

## Figures and Tables

**Figure 1 diagnostics-12-00747-f001:**
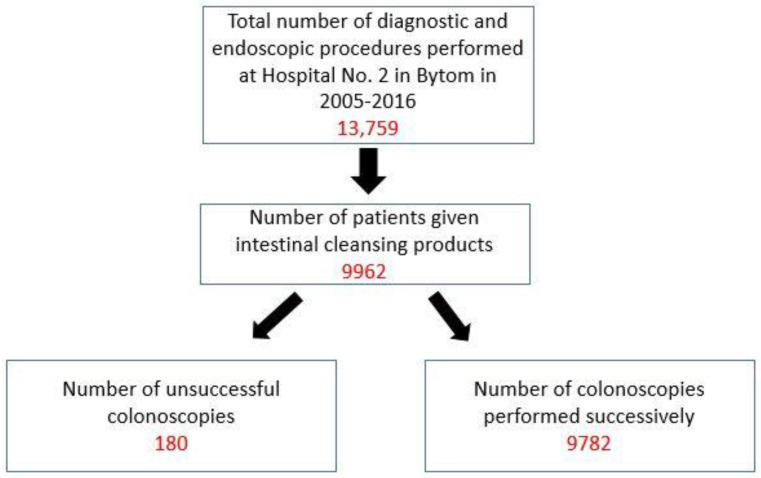
Diagram of performed and analyzed colonoscopy procedures.

**Figure 2 diagnostics-12-00747-f002:**
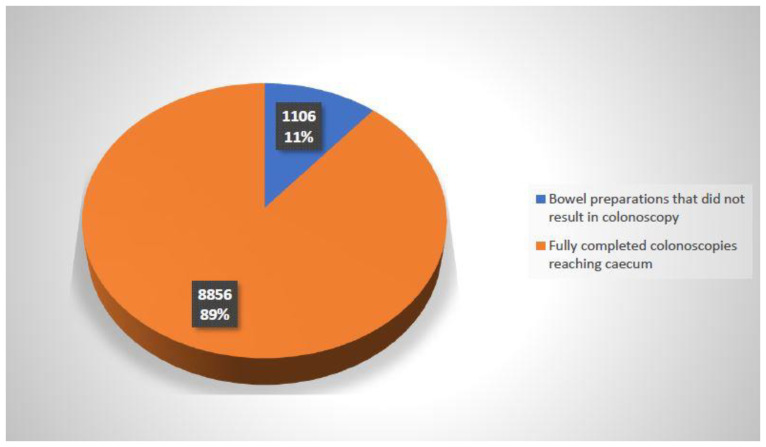
Quantity and percentage of non-fully successful bowel preparations before in relation to all of the bowel preparations among patients qualified for colonoscopy.

**Figure 3 diagnostics-12-00747-f003:**
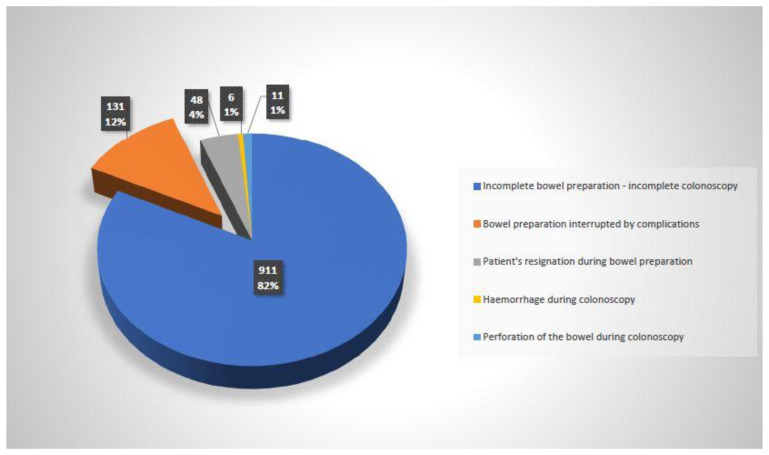
Analysis of causes of 180 unsuccessful colonoscopies, including the period of bowel preparation during direct preparation for colonoscopy.

**Figure 4 diagnostics-12-00747-f004:**
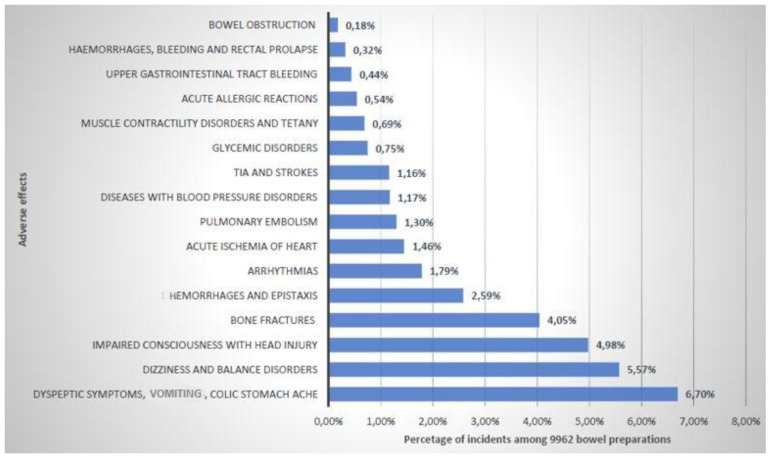
Prevalence of adverse effects occurring during the analysis of 9962 initiated bowel preparation regimens that required medical intervention at the ward and/or implementation of specialized medical procedures.

## Data Availability

Data available on request due to restrictions eg privacy or ethical.

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
