# Peer review of "Colonoscopy: Preparation and Potential Complications"

_diagnostics, 2022, doi:10.3390/diagnostics12030747_

Round 1
Reviewer 1 Report
The authors intend to report on the incidence of side effects during the bowel preparation for colonoscopy. It is an interesting topic but the manuscript has several flaws that need to be addressed:
1) The introduction is too long. The introduction needs to focus on the topic that the paper wants to address. You should write a couple of sentences as how preparation is done and what are the possible complications. You should not repeat the same data on both the introduction and discussion.
2) The methods need to be more specific. The number of colonoscopies is confusing. Was it 13759 or 9782 or 9962? Maybe a flow chart could help the readers better understand the numbers.
3) The results section starts by describing which adverse events were considered, but this has to be done in the methods. Then after describing the adverse events the authors describe the colonoscopy findings. Was this one of the aims of the study?
4) Discussion is too long and tiring for the readers to follow. It also mentions again data already written in the Introduction
5) I really find it difficult to undestand why the authors mention in the last paragraph of the Discussion the "Barrett''s esophagus" study. The manuscript is supposed to examine the side effects associated with coloconscopy preparation. What this has to do with Barrett's ???????
6) The conclusion should be one paragraph and not half or a whole page
Author Response
Dear Reviewer, thank you very much for work to improve or manscript. Attached please find, sincerely
Reviewer 2 Report
This is an interesting retroscopcive study that will give adequate information about the state of things in colonoscopy practice in Poland. A sort of clinical audit or service evaluation. However, what this document lacks is a structured, neat and clear presentation. I would have given it a fail (rejection outwith) but as I appreciate the document contains useful info for a number of readers, I recommend taking it back to the drawing board and re-present a manuscript with short, succinct and academic-level statements, not to mention a clear split of paragraphs as well. Significant restricting in required.
Author Response
Dear Reviewer, than you very much for your work to improve our manuscript, Attached plase find, Sincerely
Round 2
Reviewer 1 Report
The manuscript has been substantially improved
Two minor comments
1) In the Introduction change "sugrical colonoscopy" to "interventional colonoscopy"
2) In the Discussion delete the sentence "perhaps it is like with aviation accidents ........ "
Author Response
Dear Reviewer,
Thank you very much for your work with us. We thank for all your comments.
Sincerely

Reviewer 2 Report
Although your attempt to provide a useful collection of the procedures, in the form of a clinical audit of your practice for a decent amount of time is to be applauded, this manuscript still falls sort of presentation quality. One would argue that when you address a specialised audience (even if not GI physicians or surgeons) with an opening statement such as "colonoscopy is a method of examining the lower digestive tract..." you are in danger of losing your readership attention due to either irritation or loss of interest. Allow me to say that in 2022, anyone that opens a medical journal does not need to be reminded what a colonoscopy is. I think this rather simplistic approach is also obvious in the presentation of the results and the figures submitted. Please take some time to tighten your manuscript (follow the 2-4-4-5 rule of paragraphs, for introduction/method/results/discussion) and avoid long and boring sentences. I would love to hear about your >10 years experience (although I come from a centre with >10k colon per year) and learn from it. However, as mentioned before, neither the text presentation nor the image colour selection and presentation allow for serious reading. Never use blue in images boxes and avoid yellow arrows. They may look nice for a PP presentation but are really frustrating for a professional reader.
Having said that, if I recall the previous version, your manuscript has significantly improved in length. It simply needs to be tightened significantly more before acceptance. I wish I could spend more time dealing with your paper but this is the work that authors should do to gain the reviewers.
Author Response
Dear reviewer, Thank you very much for work with us. We very thank you for opportunity to learn more.
Sincerely

Round 3
Reviewer 2 Report
The manuscript has been improved, to the point to be considered acceptable for publication.
However, a couple of references are missing and should be included in the appropriate section of the discussion:
https://pubmed.ncbi.nlm.nih.gov/35204593/
https://pubmed.ncbi.nlm.nih.gov/34540535/